# Molecular Tetris by sequence-specific stacking of hydrogen bonding molecular clips

Hyun Lee [1] & Dongwhan Lee [1]✉

A face-to-face stacking of aromatic rings is an effective non-covalent strategy to build functional architectures, as elegantly exemplified with protein folding and polynucleotide assembly. However, weak, non-directional, and context-sensitive van der Waals forces pose a significant challenge if one wishes to construct well-organized π-stacks outside the confines of the biological matrix. To meet this design challenge, we have devised a rigid polycyclic template to create a non-collapsible void between two parallel oriented π-faces. In solution, these shape-persistent aromatic clips self-dimerize to form quadruple π-stacks, the thermodynamic stability of which is enhanced by self-complementary N–H···N hydrogen bonds, and finely regulated by the regioisomerism of the π-canopy unit. With assistance from sufficient electrostatic polarization of the π-surface and bifurcated hydrogen bonds, a small polyheterocyclic guest can effectively compete against the self-dimerization of the host to afford a triple π-stack inclusion complex. A combination of solution spectroscopic, X-ray crystallographic, and computational studies aided a detailed understanding of this cooperative vs competitive process to afford layered aromatics with extraordinary structural regularity and fidelity.

[1] Department of Chemistry, Seoul National University, 1 Gwanak-ro, Gwanak-gu, Seoul 08826, Korea. ✉email: dongwhan@snu.ac.kr

Electronically tunable columnar stacks of aromatic molecules[1,2] are key functional components in molecular devices and materials[3–9]. To define efficient pathways for charge or energy carriers, precise molecular-level control of the π–π stacking is needed[10–15]. For example, a linear molecular backbone can support regularly spaced aromatic branches that define a tight π-stack, as exemplified with comb-like polymers[16–19], hairpin turn-motifs[20–25], and multi-layer cyclophanes[26,27]. Despite intuitive appeal in design, with flexible polymers, it is not easy to control the relative orientations of the planar π-branches along the backbone axis[17]. For small molecules, the spacing between individual π-faces might be controlled precisely, but covalent construction of multi-layer molecules to achieve long-range structural ordering is a synthetically demanding task.

To overcome these challenges, supramolecular systems have been developed by using non-covalent interactions between small-molecule building blocks[28–33]. Prominent examples include aromatic guest-encapsulated molecular cages[14,34–38], through-space π-conjugated foldamers[39–41], metal coordination-induced π-stacking[42–44], and self-assembled molecular tweezers[45–49]. To control the special ordering and orientation of π-faces with high fidelity, a rational molecular design strategy is needed[50–54]. In this paper, we report the chemistry of shape-persistent molecular clips.

As schematically shown in Fig. 1, a C-shaped (or horizontal lying letter U-shaped) molecular skeleton was designed to support two parallel-oriented π-surfaces that form tight quadruple stacks upon self-dimerization. With pre-defined anti-parallel orientations between alternating stacks and N–H⋯N hydrogen bonds in the middle[55,56], only a *a–b–b–a* type π-sequence, not an isomeric *a–a–b–b* or *b–a–a–b*, is allowed for the quadruple layer[57]. The regularity in stacking orientation[58,59] and stacking sequence[32,57,59] facilitate the self-association of quadruple π-stacks into an infinite columnar π-stacks in the solid state. We also demonstrate that the quadruple π-stacks can disassemble into individual π-clips to encapsulate a small polyheterocyclic molecule, thereby affording a guest-intercalated *a–c–b* type triple π-stacks. Such structure- and context-dependent Tetris-like

assembly is a testament to the importance of individually weak yet collectively strong non-covalent interactions, the details of which are provided in the following sections.

## Results and discussion

**Design principles: shape-matching cavities and complementary hydrogen bonds.** To build rigid clip-like molecules that self-assemble into anti-parallel π-stacks (Fig. 1), a spiro junction was devised to support a sharp 90° turn between the cyclic ketone-fused indole and isobenzimidazole. As shown in Fig. 1, such a molecular scaffold orients two different π-surfaces *a* and *b* in a parallel fashion, while leaving a non-collapsible void in between. In addition to functioning as a rigid structural support, the indole unit provides an N–H group as a hydrogen-bonding donor (HBD). The intermolecular N–H⋯N hydrogen bonds ensure that only a *a–b–b–a* type π-sequence takes full advantage of the shape complementarity and the maximum number of hydrogen bonds.

Using this general structural platform, a homologous set of molecular clips **C-NI**, **C-P1**, **C-P2**, and **C-P4** were designed. While sharing the common C-shaped skeleton, these molecules differ in the chemical composition (**C-NI** vs. **C-P**$n$, where $n = 1$, 2, and 4) and regiochemistry (**C-P1**, **C-P2**, and **C-P4**) of the upper canopy unit (Fig. 2). The conformationally rigid nature of such π-clip motif is reflected in the presence of only one rotational bond along the entire molecular backbone: $C_{indole}–N_{imide}$ for **C-NI**; $C_{indole}–C_{pyrene}$ for **C-P**$n$ ($n = 1$, 2, and 4).

**Synthesis of π-clips.** The synthesis of the upper part of the π-clip required direct C–C cross-coupling reactions of cyclic ketone-fused bromoindole **1** with different regioisomers of pyrene boronic ester/acids to prepare **2** (Fig. 2). Alternatively, the bromo group of **1** was converted to azide, and subsequently reduced to the amine intermediate **3**, which was condensed with 1,8-naphthalic anhydride to afford **4**. Each of the ketone intermediates (**2** or **4**) was subjected to a condensation reaction with the common precursor **5** to afford the corresponding aminals **6**. Unlike ketals, aminals are prone to hydrolysis back to ketone and amine[60,61];

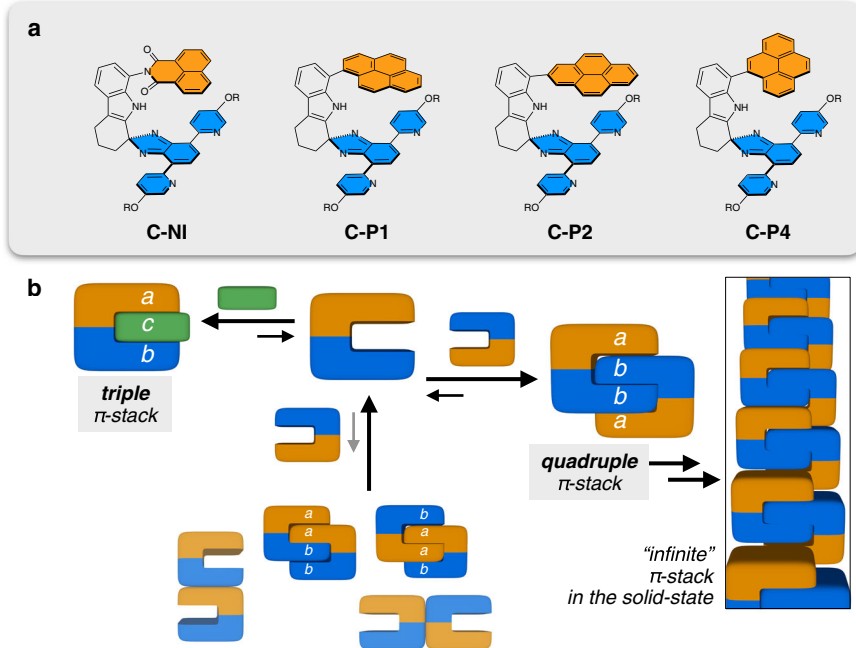

**Fig. 1 Design principles of π-clips and sequence-specific assembly into π-stacks. a** Chemical structures of a homologous series of π-clip molecules **C-NI**, **C-P1**, **C-P2**, and **C-P4**. **b** Schematic representations of molecular assembly into triple π-stack, quadruple π-stack, and infinite π-column.

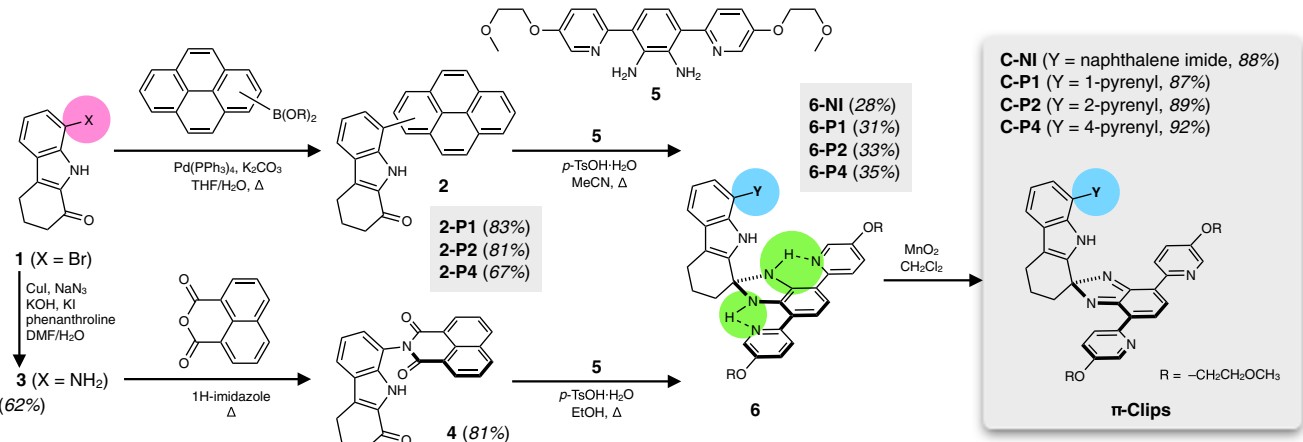

**Fig. 2 Synthesis of π-clips.** The modular construction of **C-NI** and regioisomeric **C-P***n* (*n* = 1, 2, and 3) series π-clip molecules. THF tetrahydrofuran, DMF *N,N*-dimethylformamide, *p*-TsOH *para*-toluenesulfonic acid, Me methyl, Et ethyl.

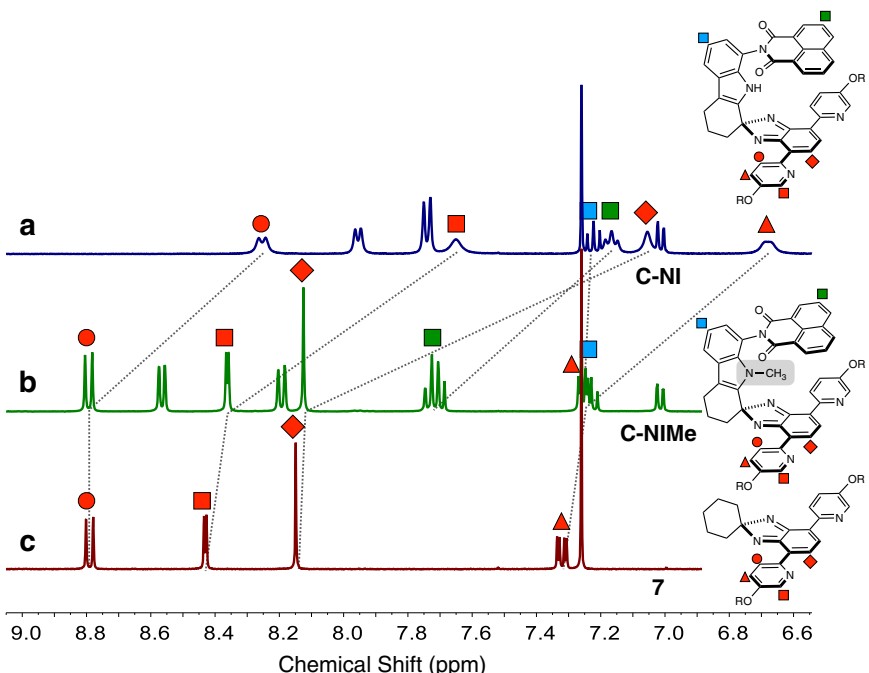

**Fig. 3 Spectroscopic signatures of intermolecular association.** Partial $^1$H NMR (400 MHz) spectra of **a C-NI** (9.2 mM), **b C-NIMe** (13 mM), and **c 7** (9.2 mM) in CDCl$_3$ at $T$ = 298 K. NMR nuclear magnetic resonance.

few isobenzimidazole are thus known to have aromatic substituents at the quaternary carbon center[62,63]. For **6**, two intramolecular N–H⋯N hydrogen bonds (green circles, Fig. 2), as evidenced by significantly downfield-shifted N$_{aminal}$–H proton resonances at $\delta$ = 7.57–7.32 ppm, seem to facilitate the condensation reaction by shifting the equilibrium to the product side. An oxidation reaction with MnO$_2$ cleanly converted **6** to the final products. High modularity in our synthetic design allows for the construction of four different π-clips **C-NI**, **C-P1**, **C-P2**, and **C-P4**, all sharing the same L-shaped lower skeleton, but differing in the upper canopy unit that defines the overall C-shaped molecular architecture (see Supplementary Methods for details on synthesis and characterization).

**Self-assembly in the solution and solid states.** An initial indication of the self-dimerization of π-clips (Fig. 1) came from comparative $^1$H NMR spectroscopic studies of **C-NI**, its $N$-

methylated derivative **C-NIMe**, and the substructure model **7** (Fig. 3). While similar aromatic proton resonances were observed for **C-NIMe** and **7** (Fig. 3b, c), **C-NI** displayed dramatic upfield shifts ($\Delta\delta$ = 0.54–1.07 ppm) of the naphthyl, pyridyl, and isobenzimidazolyl C–H protons (Fig. 3a). In contrast, the indole C–H proton resonances remained essentially unchanged.

A subsequent single-crystal X-ray diffraction (SC-XRD) study on **C-NI**, grown by vapor diffusion of Et$_2$O into a CHCl$_3$ solution at r.t., revealed the formation of dimeric structure [**C-NI**]$_2$ (Figs. 1 and 4; Supplementary Fig. 1a). Upon self-association of the two π-clips, a quadruple π-stack is created, with two symmetrically disposed N$_{indole}$–H⋯N$_{pyridyl}$ hydrogen bonds ($d_{N⋯N}$ = 2.944(2) Å) buried deep inside the hydrophobic cavity (Fig. 4b). When viewed along the vertical direction (Fig. 4c and d), the quadruple π-stack is arranged in the sequence of naphthaleneimide–isobenzimidazole–isobenzimidazole–naphthaleneimide, i.e., *a–b–b–a* (Fig. 1). The interplanar distances of 3.126(1) Å and 3.240(2) Å, determined for the

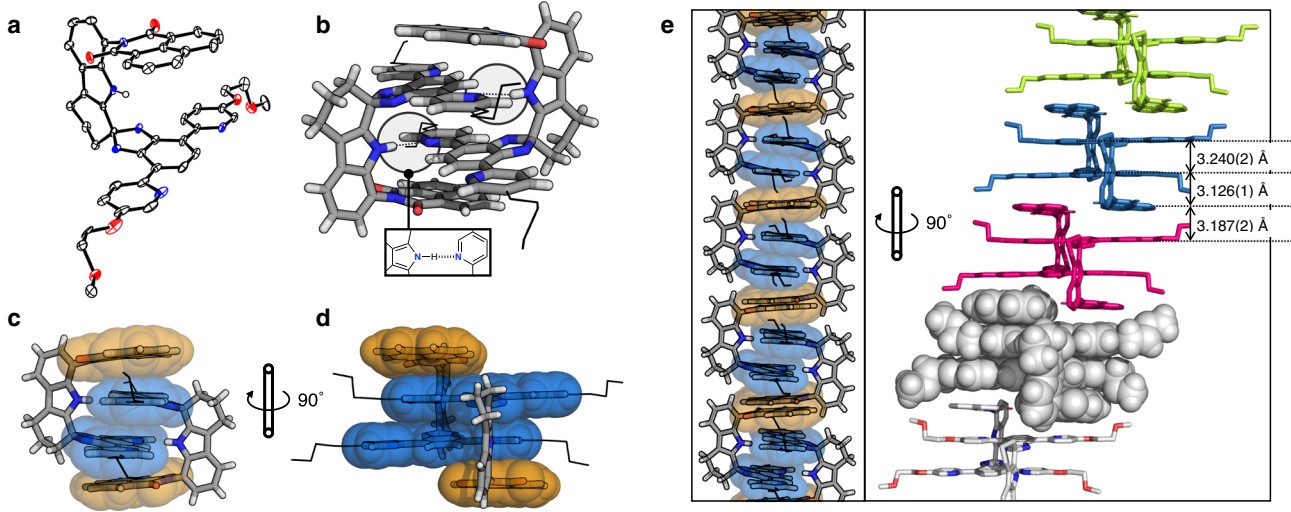

**Fig. 4 Self-dimerization and hierarchical self-assembly.** X-ray structures of **C-NI** as **a** ORTEP diagram with thermal ellipsoids at the 50% probability level. Hydrogen atoms have been omitted for clarity, except for the indole N–H group participating in the intermolecular N–H···N$_{pyridyl}$ hydrogen bond shown in (**b**). Capped-stick representations **b**–**d** are constructed from crystallographically determined atomic coordinates of [**C-NI**]$_2$ viewed from different perspectives. The ether chains are simplified as wireframes; van der Waals surfaces are overlaid on the aromatic groups constituting the quadruple π-stacks with color schemes shown in Fig. 1. **e** Capped-stick and space-filling models of an infinite vertical stack of {[**C-NI**]$_2$}$_∞$; each [**C-NI**]$_2$ dimer is color-coded differently to denote the repeating unit and highlight inter-dimer shape complementarity. ORTEP Oak Ridge thermal ellipsoid plot.

naphthaleneimide···isobenzimidazole (= $a$···$b$) and isobenzimidazole···isobenzimidazole (= $b$···$b$) contacts, respectively (Fig. 4e, right), are indicative of intimate π–π stacking. These values were determined by measuring the distance between the mean plane centroid of naphthaleneimide and the mean plane centroid of pyridine–isobenzimidazole–pyridine unit (for $a$···$b$), and the mean centroids of the two stacked (and thus facing) pyridine–isobenzimidazole–pyridine triads (for $b$···$b$).

In the solid-state, dimeric [**C-NI**]$_2$ stack further to afford an infinite one-dimensional column with an inter-dimer π–π stacking distance of 3.187(2) Å (Figs. 1 and 4e, right). The shape complementarity between adjacent [**C-NI**]$_2$ units leaves no gaps in the wire-like arrangement of the vertical π-stack. The critical functional role of the N$_{indole}$–H···N$_{pyridyl}$ hydrogen bonds in the dimer assembly is supported by the X-ray structure of **C-NIMe** (Supplementary Figs. 1e and 2). Here, methylation of the indole N–H group of the parent **C-NI** molecule removes the hydrogen bond and sterically inhibits access to the cavity. As a result, no dimerization occurs, which is consistent with the $^1$H NMR spectroscopic data (Fig. 3b). Unlike **C-NI** (Supplementary Fig. 3a and Supplementary Fig. 38), the MALDI-TOF mass spectrum of **C-NIMe** (Supplementary Fig. 3e) lacks the molecular ion peak corresponding to the dimeric [2M + H]$^+$ species.

With the structural evidence for self-association obtained by X-ray crystallography, we revisited the NMR spectra of **C-NI** to fully understand the unusually up-field shifted aromatic resonances (Fig. 3a). The 2D-ROESY $^1$H NMR spectrum of **C-NI** (Supplementary Fig. 4) also revealed strong cross-peaks that reflect through-space interactions of isobenzimidazole with indole; isobenzimidazole with naphthaleneimide; naphthaleneimide with methoxyethyl ether chain. Such spatial proximity could only be achieved by self-dimerization (Fig. 4b), which maximizes van der Waals (vdW) contacts and the number of hydrogen bonds.

**Fluxional motions of the quadruple π-stack**. The simple $^1$H NMR spectral pattern of [**C-NI**]$_2$ at r.t. (Fig. 3a) requires the

presence of a mirror plane bisecting the dimeric structure. This prediction, however, is not consistent with the X-ray structure of [**C-NI**]$_2$ with offset stacked dimer of $C_i$ local symmetry (Fig. 4b, d). To understand these seemingly conflicting experimental observations, we proceeded to carry out variable-temperature (VT) $^1$H NMR spectroscopic studies (Fig. 5). Upon lowering the temperature from $T = 25$ °C to −70 °C, the proton resonances of [**C-NI**]$_2$ in CD$_2$Cl$_2$ underwent gradual broadening and eventual splitting (Fig. 5a), which implicated slower exchange of the magnetic environments by fluxional motions of the quadruple π-stack (Fig. 5b; Supplementary Movie). Under similar conditions, the $^1$H NMR spectra of the $N$-methylated derivative **C-NIMe** showed no such temperature dependence (Supplementary Fig. 5). As shown in Fig. 4b and d, only one of the two pyridyl nitrogen atoms of each **C-NI** molecule within [**C-NI**]$_2$ can engage in the intra-dimer N$_{indole}$–H···N$_{pyridyl}$ hydrogen bonds breaking the mirror symmetry. At sufficiently low temperatures, the lateral sliding motion (Fig. 5b) to exchange the hydrogen-bonding partners becomes sufficiently slow to create inequivalent magnetic environments. Indeed, the 2D-COSY $^1$H NMR spectrum measured at $T = −70$ °C confirmed the $C_i$-symmetric structure of [**C-NI**]$_2$ with the fluxional motions frozen out (Supplementary Fig. 6), which is consistent with the X-ray structure. The offset stacked geometry of [**C-NI**]$_2$ (Fig. 4b, d) is thus an intrinsic supramolecular property, not an artifact of crystal packing.

From the coalescence temperature $T_c = 253$ K and $\Delta\nu = 510.25$ Hz (Fig. 5a), an energy barrier of $\Delta G^\ddagger = 11.2$ kcal mol$^{-1}$ could be estimated for the lateral sliding motion shown in Fig. 5b. Considering that the sliding motion depicted in Fig. 5b should overcome the steric clash between the N$_{indole}$–H group and the C–H bonds of the isobenzimidazole ring in the transition state, this value could be considered the upper limit of the ground-state stabilizing effect of the hydrogen bonds.

**Structure-dependent energetics of dimerization**. In the solution phase, the quadruple π-stack of [**C-NI**]$_2$ is thermodynamically robust. No spectral changes were observed when the sample

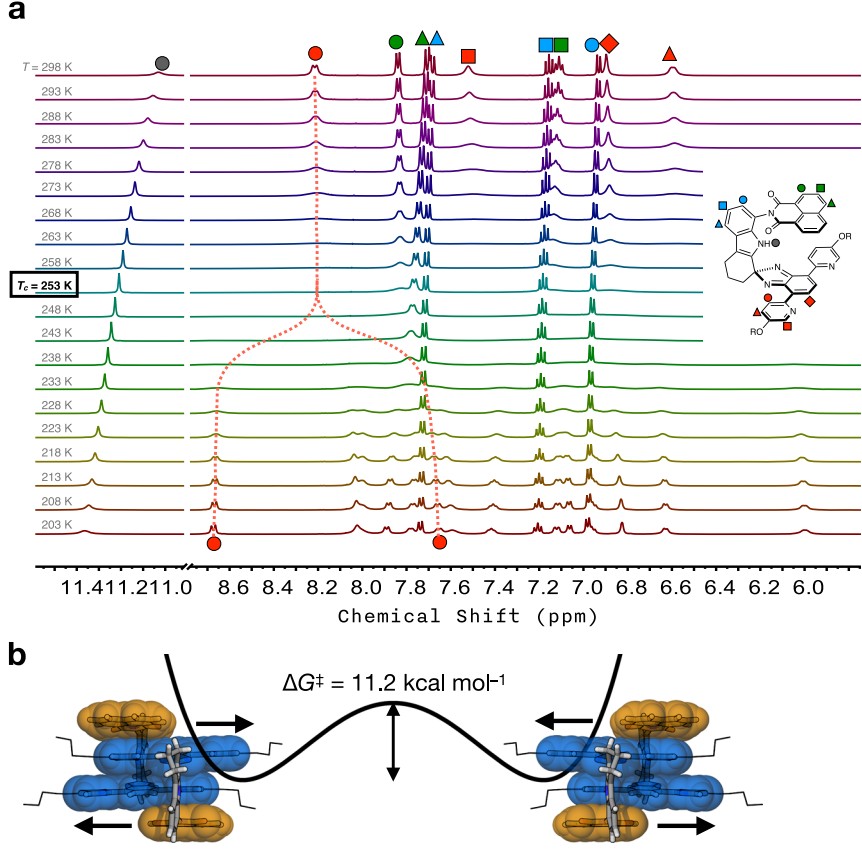

**Fig. 5 Fluxional motions of quadruple π-stack. a** Variable-temperature (VT) $^1$H NMR (500 MHz) spectra of **C-NI** in $CD_2Cl_2$ (9.5 mM) obtained at $T = 203$–298 K. The coalescence temperature $T_c = 253$ K was determined with the pyridine proton resonance (annotated as a red circle in the chemical structure shown next). **b** Schematic representation of the lateral sliding motion and the energy barrier. NMR nuclear magnetic resonance.

concentration was varied from 13.2 mM to 1.4 mM in CDCl$_3$ at r.t. (Fig. 6b). In stark contrast, the 2-pyrenyl-substituted **C-P2** (Figs. 1 and 2; Supplementary Fig. 1c) displayed significant downfield shifts in the π–π stacked region upon sample dilution (Fig. 6c). To better understand this structure-dependent solution behavior, regioisomeric **C-P1** and **C-P4** (Figs. 1 and 2; Supplementary Fig. 1b, d) were prepared for comparative studies. Similar to **C-P2**, both **C-P1** and **C-P4** also produced self-dimerized structures upon crystallization (Figs. 1 and 6a). Density functional theory (DFT) energy-minimized structures of the π-clips overlap nicely with the crystallographically determined structures (Supplementary Fig. 7), which demonstrates the rigid non-collapsible nature of the C-shaped structural preorganization for lock-and-key type self-association.

Despite what appeared to be minor differences in the attachment position of the pyrenyl canopy, however, these three isomeric quadruple π-stacks showed strikingly different assembly behavior. Using the pyridine resonances as a spectroscopic handle, concentration-dependent $^1$H NMR spectra were obtained at $T = 298$ K. Using monomer–dimer isotherm fitting (Supplementary Note 1), the self-dimerization constants ($K_{dim}$) were determined as 8.1 ($\pm2.3$) $\times 10^2$ M$^{-1}$ for **C-P1**, and 1.9 ($\pm0.3$) $\times 10^2$ M$^{-1}$ for **C-P2** in CDCl$_3$ (Supplementary Figs. 8 and 9)[60,61]. In more polar $CD_2Cl_2$ solvent, a slightly higher $K_{dim}$ value of 3.4 ($\pm0.4$) $\times 10^2$ M$^{-1}$ was obtained for **C-P2** (Supplementary Fig. 17). The validity of the simple non-linear regression was further checked by global fitting analysis using additional proton resonances showing concentration dependence, which furnished comparable $K_{dim}$ values (Supplementary Figs. 12–15). For **C-P4**, a severe peak broadening prevented the determination of the $K_{dim}$

value under the experimental conditions that we employed (Supplementary Fig. 10). The MALDI-TOF spectra of **C-P1**, **C-P2**, and **C-P4** (Supplementary Fig. 3b–d) consistently showed $m/z$ peaks corresponding to the dimeric $[2 M + H]^+$ species.

A four-fold enhancement in the stability of $[\textbf{C-P1}]_2$ over $[\textbf{C-P2}]_2$ is quite unexpected since they differ only in the substitution position at the pyrene ring (Fig. 6a). On an energy scale, this translates to ca. 0.8 kcal mol$^{-1}$ in the preferential stabilization. An analysis of the Connolly surfaces using different probe radii (0.8–2.5 Å) revealed that the solvent-accessible surface areas are essentially identical for monomeric **C-P1** and **C-P2** (Supplementary Fig. 19). On the other hand, the exposed surface of dimeric $[\textbf{C-P1}]_2$ is always smaller than that of $[\textbf{C-P2}]_2$ (Supplementary Fig. 19), indicating more extensive intradimer vdW contacts and thus tighter association of the former than the latter. A subtle difference in the regioisomerism thus translates to measurable changes in self-association when it becomes an integral part of the π-clip. With the lowest dimerization constant ($K_{dim} = 1.9$ ($\pm0.3$) $\times 10^2$ M$^{-1}$), the $^1$H NMR spectrum of a dilute ($2.4 \times 10^{-2}$ mM) sample of **C-P2** is similar to that of the non-stacking model **7** and the $N$-methylated derivative **C-P2Me** (Fig. 6c, and Supplementary Figs. 20 and 39), when the proton resonances of the pyridyl–isobenzimidazole regions are compared.

Unlike **C-P1** and **C-P2**, the essentially concentration-independent $^1$H NMR spectra of **C-NI** (Fig. 6b) made it difficult to analyze the monomer–dimer equilibrium within the concentration range that does not suffer from line broadening. To approximate the free (= non-stacked) **C-NI**, we instead used the pyridine resonance of the model compound **C-NIMe** (Fig. 3b,

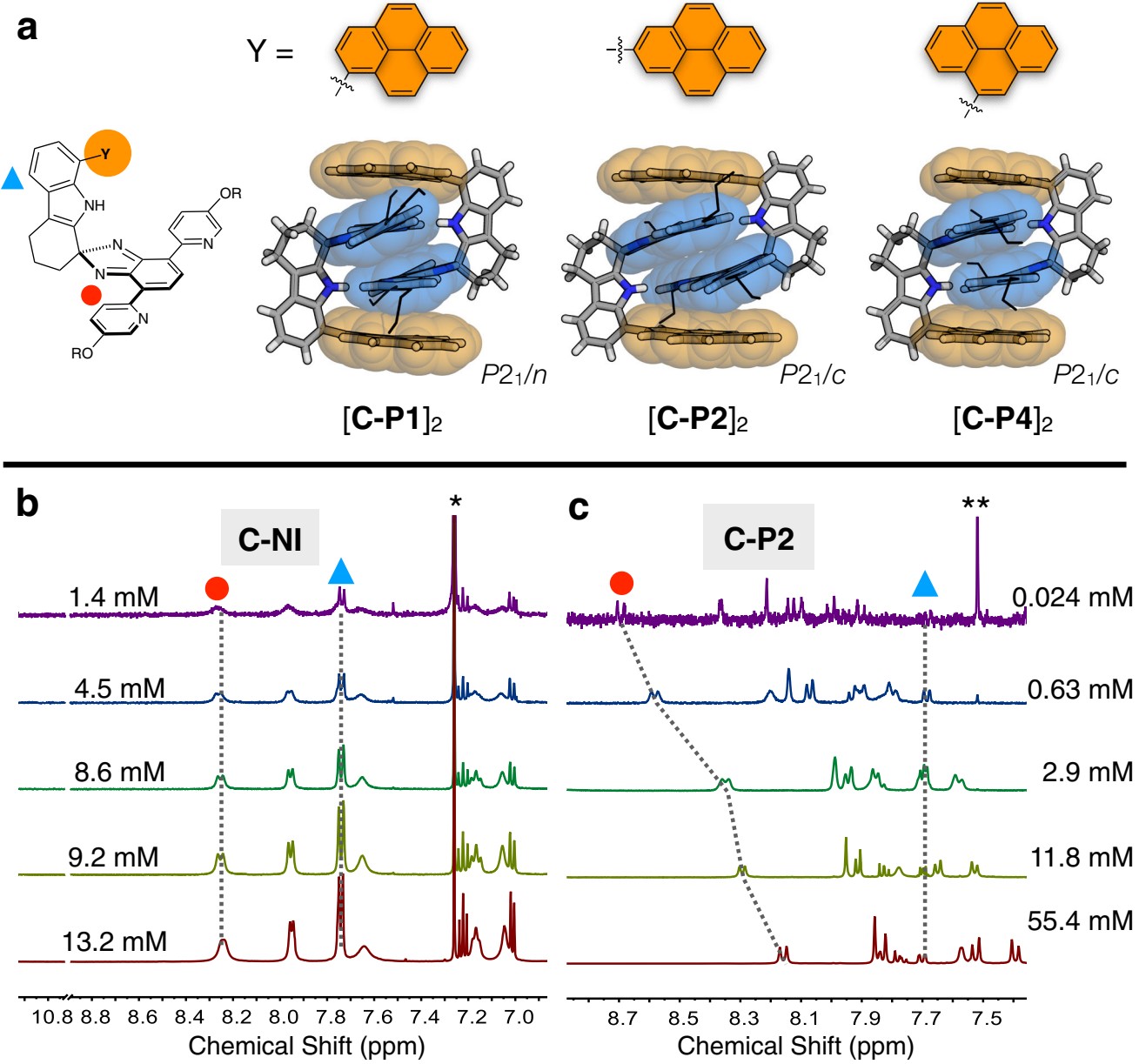

**Fig. 6 X-ray structures of regioisomeric dimers and structure-dependent monomer–dimer equilibrium. a** Capped-stick representation of the X-ray structures of [**C-P***n*]$_2$ (*n* = 1, 2, and 4). Partial $^1$H NMR (400 MHz, *T* = 298 K) spectra of **b C-NI** and **c C-P2** were obtained by varying the concentration of CDCl$_3$ solution samples. Proton resonances are labeled with the symbols denoted in the chemical structure shown in **a**. The symbols * and ** denote residual solvent and satellite peaks, respectively. NMR nuclear magnetic resonance.

$\delta$ = 8.793 ppm) to fit the isotherm (Supplementary Fig. 11). The estimated value of $K_{\text{dim}}$ = 1.3 ($\pm$0.8) $\times$ 10$^5$ M$^{-1}$ is sufficiently large so that the solution population is dominated by the dimeric [**C-NI**]$_2$. The variable temperature (VT) (Fig. 5) and 2D ROESY (Supplementary Fig. 4) NMR spectroscopic, and MALDI-TOF mass spectrometric (Supplementary Fig. 3a) data corroborate this notion. For a 5 mM solution sample of **C-NI** in CDCl$_3$ at r.t., the mole fraction of monomeric **C-NI** would be as low as 4%. A further dilution of the NMR sample (down to 0.09 mM) and overnight scanning barely shifted the pyridine resonance (Supplementary Fig. 16), suggesting that the $K_{\text{dim}}$ value of 1.3 ($\pm$0.8) $\times$ 10$^5$ M$^{-1}$ is the lower limit of the self-dimerization affinity of **C-NI**. Similar to the situation of **C-P2** showing an enhanced dimerization propensity in more polar CD$_2$Cl$_2$ (Supplementary Figs. 9 and 17), the $K_{\text{dim}}$ value of **C-NI** also

increased to $K_{\text{dim}}$ = 3.3 ($\pm$0.9) $\times$ 10$^5$ M$^{-1}$ in CD$_2$Cl$_2$ (Supplementary Figs. 11 and 18).

The strong self-dimerization propensity of **C-NI** was further manifested in comparative studies with **C-P2**. In DMSO-$d_6$ at *T* = 25 °C, the $^1$H NMR spectrum of **C-P2** revealed the co-existence of monomer and dimer (Supplementary Fig. 21), whereas **C-NI** largely maintained its dimeric structure (Supplementary Fig. 23). Upon increasing the temperature, however, a dramatic sharpening of the peaks occurred for **C-NI** (Supplementary Fig. 25a). Notably, the spectral pattern of **C-NI** in DMSO-$d_6$ observed at *T* = 80 °C is very similar to that of **C-NIMe** in CDCl$_3$ at *T* = 25 °C (Supplementary Fig. 26), suggesting that [**C-NI**]$_2$ could dissociate into monomeric **C-NI** in hydrogen-bonding solvent at high temperatures. In stark contrast, only slight downfield shifts were observed in

1,2-dichloroethane-$d_4$ even at elevated temperatures (Supplementary Fig. 25b). Dimeric $[\textbf{C-NI}]_2$ thus seems to dominate the solution population in non-hydrogen bonding solvent at the entire temperature range.

DFT computational studies showed a large difference in the molecular dipole moment of 3.02 D for **C-NI** vs. 1.54 D for **C-P2** (Supplementary Fig. 27), suggesting that the two antiparallel oriented π-clips within $[\textbf{C-NI}]_2$ would benefit from stronger dipole–dipole interaction for enhanced stability. Moreover, the molecular electrostatic potential (MEP) map revealed the electron-deficient nature of naphthaleneimide (Supplementary Fig. 27), the upper canopy unit of **C-NI**, which always stacks with the bottom isobenzimidazole layer of the other molecule upon self-dimerization to $[\textbf{C-NI}]_2$. A donor–acceptor type electrostatic interaction would thus contribute further to the stability of the quadruple π-stacks of $[\textbf{C-NI}]_2$. With a less electronic distinction between the pyrene canopy and isobenzimidazole bottom, $[\textbf{C-P2}]_2$ seems to benefit less from such secondary effects. In pyridine-$d_5$, however, dimer disassembly was observed for both **C-NI** and **C-P2** (Supplementary Figs. 22 and 24), indicating that even electrostatically augmented vdW contacts are not sufficient to outcompete the loss of intramolecular hydrogen bonds that hold the quadruple π-stack. $^1$H DOSY measurements on **C-NI** and **C-P$n$** ($n = 1, 2,$ and 4) in CDCl$_3$ provided diffusion coefficients of 8.80–9.55 × 10$^{-10}$ m$^2$ s$^{-1}$, which are smaller than that of monomeric **C-NIMe** (1.13 × 10$^{-9}$ m$^2$ s$^{-1}$), but indicate no significant inter-dimer interactions in the solution state (Supplementary Fig. 28). In support of this notion, their molar absorptivities are also independent of the sample concentration (Supplementary Fig. 29).

**Aromatic intercalation to triple π-stacks.** To explore the practical utility of π-stacks beyond simple self-association, we decided to use the shape-persistent void for guest intercalation. For the host–guest complexation to occur in solution, the ideal guest molecule needs to (i) have a flat and large π-surface to fit inside the interplanar void, and (ii) form hydrogen bond(s) with the indole N–H group inside the cavity. Among various small molecules (see Supplementary Figs. 35 and 36 for a complete list) screened in the exploratory studies, 1,10-phenanthroline-5,6-dione (PHD) and **C-P2** produced a 1:1 host–guest complex as diffraction quality crystals. Subsequent SC-XRD studies revealed the formation of an inclusion complex **C-P2⊃PHD** with tight π–π stacking distances of 3.3265(18) Å and 3.8182(17) Å, and dissymmetric bifurcated hydrogen bonds with $d_{\text{N–H···N}} = 2.947(5)$ Å and 3.442(5) Å (Fig. 7 and Supplementary Fig. 1f).

In CDCl$_3$ at r.t., a mixture of **C-P2** and PHD produced large chemical shifts in the aromatic proton resonances of both **C-P2** and PHD (Supplementary Fig. 30). The proton resonances associated with **C-P2** reflect solution equilibrium involving at least

three different species, **C-P2**, $[\textbf{C-P2}]_2$, and **C-P2⊃PHD**, which is further complicated by their rapid exchange to produce weight-averaged broad signals that could not be deconvoluted into individual components by numerical fitting alone. We thus decided to employ the simpler $^1$H NMR patterns of the guest PHD, which exists as either host-entrapped **C-P2⊃PHD** or unbound guest PHD[64]. Using the C–H proton resonances of PHD as a spectroscopic handle, a global non-linear regression (see Supplementary Note 2 for details on the numerical fitting model) produced a host–guest complexation constant of $K_{\text{HG}} = 2.0\ (\pm 0.1) \times 10^2$ M$^{-1}$ (Fig. 8a and Supplementary Fig. 30)[65]. Under similar conditions, a comparable value of $K_{\text{HG}} = 2.6\ (\pm 0.1) \times 10^2$ M$^{-1}$ was estimated for **C-P1⊃PHD** (Supplementary Fig. 31); titration studies with **C-P4**, however, were hampered by significant line broadening and signal overlap.

A Job plot analysis (Supplementary Fig. 33) further established a 1:1 binding stoichiometry between **C-P2** and PHD in solution. An increasing concentration of the PHD also elicited a gradual downfield shift of the pyridyl C–H and an upfield shift of the pyrenyl C–H proton resonances of **C-P2** (Supplementary Fig. 34), which is consistent with the X-ray structure of the inclusion complex **C-P2⊃PHD** shown in Fig. 7. To investigate the geometric and electronic factors dictating guest intercalation, $^1$H NMR spectra of **C-P2** were obtained in the presence of various aromatic molecules of structural resemblance to PHD, including 9,10-phenanthrenequinone (PHQ), phenanthrene, 1,10-phenanthroline (phen), and 2,2′-bipyridine (bpy). As shown in Fig. 9a and Supplementary Fig. 35, significant changes in chemical shifts were observed only for PHD. Nonpolar polyaromatic hydrocarbons (PAHs) of flat (e.g., perylene, triphenylene, and pyrene) or curved

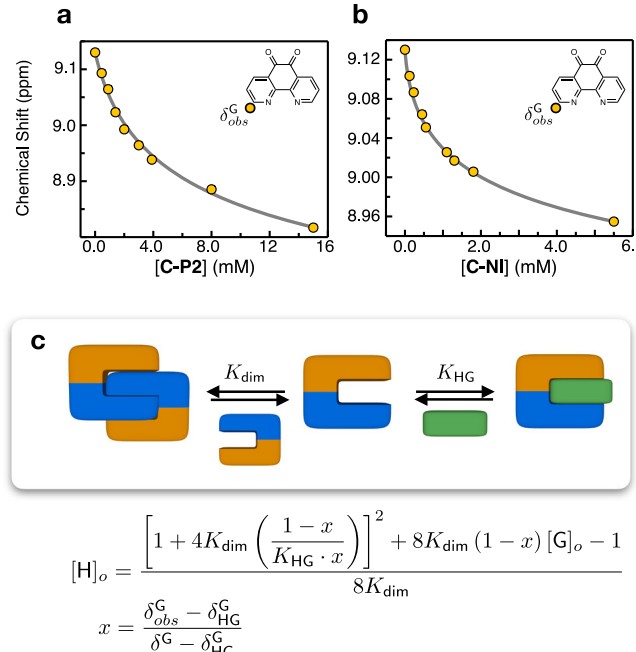

$$[\text{H}]_o = \frac{\left[1 + 4K_{\text{dim}}\left(\dfrac{1-x}{K_{\text{HG}} \cdot x}\right)\right]^2 + 8K_{\text{dim}}(1-x)[\text{G}]_o - 1}{8K_{\text{dim}}}$$

$$x = \frac{\delta^{\text{G}}_{obs} - \delta^{\text{G}}_{\text{HG}}}{\delta^{\text{G}} - \delta^{\text{G}}_{\text{HG}}}$$

**Fig. 8 Competitive π-stacking: self-association vs. host–guest complexation.** Representative changes in the chemical shift of the guest ($\delta^{\text{G}}_{obs}$, annotated by a yellow circle in the chemical structure; [PHD]$_0$ = 0.45 mM) in CDCl$_3$ during titration with either **a C-P2** or **b C-NI** at $T$ = 298 K. The overlaid gray lines are theoretical fits that take into account both self-association ($K_{\text{dim}}$) and guest encapsulation ($K_{\text{HG}}$) of the host, as described in **c**: $\delta^{\text{G}}_{\text{HG}}$ = chemical shift of the host–guest complex (either **C-P2⊃PHD** or **C-NI⊃PHD**); $\delta^{\text{G}}$ = chemical shift of the unbound guest. For each case, $K_{\text{HG}}$ was determined by global fitting (see Supplementary Figs. 30 and 32). PHD 1,10-phenanthroline-5,6-dione.

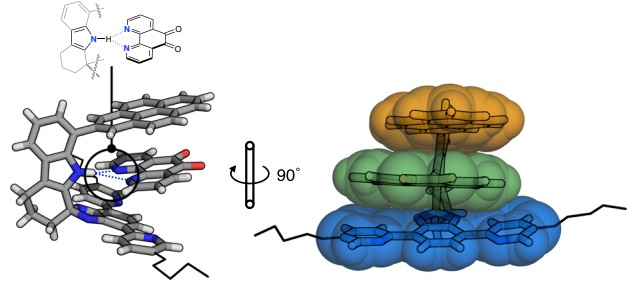

**Fig. 7 Host–guest complexation to build a triple π-stack.** Capped-stick representation of the X-ray structure of **C-P2⊃PHD** with vdW surfaces overlaid on the stacked aromatic regions, and ether chains simplified as wireframes. PHD 1,10-phenanthroline-5,6-dione, vdW van der Waals.

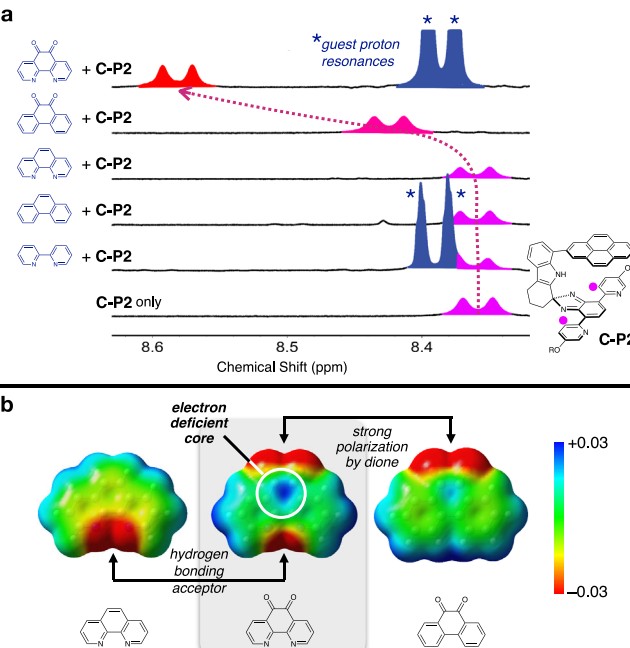

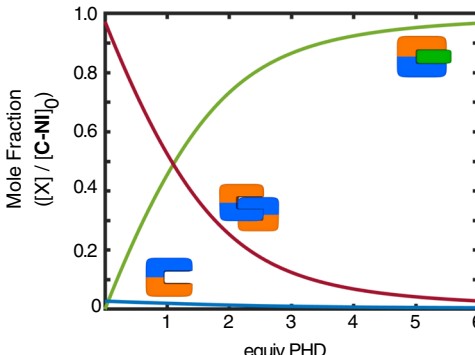

**Fig. 10 Molecular reshuffling.** Speciation plot showing changes in the mole fractions of individual species originating from the π-clip (unbound **C-NI**, blue line; dimeric [**C-NI**]₂, red line; host–guest complex **C-NI⊃PHD**, green line) with the supply of the guest PHD. [**C-NI**]₀ = 5.0 mM; see Fig. 8 for the $K_{dim}$ and $K_{HG}$ values used in the numerical simulation. PHD 1,10-phenanthroline-5,6-dione.

**Fig. 9 Selective intercalation of π-conjugated guest. a** Partial ¹H NMR spectra (400 MHz) of **C-P2** obtained in the presence of various poly(hetero)aromatic guests G in CDCl₃ at $T$ = 298 K ([**C-P2**] = 4.0 mM; [G] = 16.0 mM). The pyridyl proton (denoted with purple circles in the chemical structure of **C-P2**) resonances are shaded with purple to red. **b** MEP maps of phen, PHD, and PHQ, calculated at the B3LYP-D3/6-31 g(d,p) level of theory. NMR nuclear magnetic resonance, MEP molecular electrostatic potential, phen phenanthrene, PHD 1,10-phenanthroline-5,6-dione, PHQ phenanthrenequinone.

geometry (e.g., corannulene) were not recognized by **C-P2**, either (Supplementary Fig. 36). The molecular electrostatic potential (MEP) maps revealed the electron-deficient nature of the π-surface of PHD relative to PHQ (Fig. 9b). Apparently, the strong complexation of PHD by **C-P2** is a synergistic effect of N–H···N hydrogen bonds and donor–acceptor–donor type electrostatic π–π stacking working in concert.

Under similar conditions, the addition of PHD to [**C-NI**]₂ also induced large shifts in the aromatic region of the ¹H NMR spectra (Supplementary Figs. 32 and 37). Using global non-linear regression, the binding isotherms were fitted with the host–guest complexation constant of $K_{HG}$ = 8.3 (±0.3) × 10³ M⁻¹ (Fig. 8b and Supplementary Fig. 32). The higher binding affinity of **C-NI⊃PHD** than **C-P2⊃PHD** might be ascribed to the larger dipole (Supplementary Fig. 27) of **C-NI** ($\mu$ = 3.02 D) relative to **C-P2** ($\mu$ = 1.54 D), thus engendering stronger antiparallel dipole–dipole interaction with entrapped PHD ($\mu$ = 2.61 D). A competitive encapsulation of the aromatic guest thus turns $a$–$b$–$b$–$a$ type quadruple π-stacks into $a$–$c$–$b$ type triple π-stacks (Fig. 1). For example, a solution of [**C-NI**]₀ = 5.0 mM at r.t. would be mainly populated by self-dimerized [**C-NI**]₂ (96%) and only a small amount of free **C-NI** (4%) (*vide supra*). The addition of the PHD guest (6 equiv) to this system reshuffles the π-stacks, from which the triple-decker **C-NI⊃PHD** would emerge as the dominant species (97%), equilibrating with only tiny fractions of the quadruple-decker [**C-NI**]₂ (2%) and unbound **C-NI** (1%). This molecular redistribution process can be graphically represented by the speciation plot in Fig. 10, which shows how the solution population is shifted from quadruple π-stacks to triple π-stacks with an increasing amount of the exogenously added guest (see Supplementary Note 3 for details on the numerical simulation).

## Conclusions

Existing molecular clips exploit rigid backbones, such as bridged fused rings[64,66], glycoluril bicycles[67–69], or annulated polyarenes[49,53,70] to create non-collapsible voids. For these systems, the molecular skeleton's curved shape[64,66] or flipping[67–69]/rotating[53,70] motion is less ideal for the sequence-specific spatial organization of multiple π-stacks. We have developed shape-persistent molecular clips. With a non-collapsible void defined by two facing flat surfaces and a rigid spacer, these molecules can self-dimerize to build quadruple stacks, or intercalate an aromatic guest to afford triple stacks.

Key to the success of our approach is a combination of shape complementarity and strategic placement of hydrogen bonds that synergistically reinforce otherwise weak and non-directional dispersive intermolecular forces. The general applicability of this design concept was demonstrated with a series of π-clips that share a common L-shaped backbone but differ in the upper canopy part's chemical structure, which profoundly impacts the thermodynamics of self-assembly and selective intercalation of hydrogen-bonding polyaromatic molecule with electronically polarized π-surfaces. Efforts are currently underway in our laboratory to expand the scope of this chemistry and find practical applications of the π-clips for sensing and switching.

## Methods

Synthetic procedures and characterization of π-clips reported in this work are provided in the Supplementary Methods.

**Physical measurements.** ¹H NMR and ¹³C NMR spectra were recorded on a 400 MHz Agilent 400-MR DD2 Magnetic Resonance System, a 500 MHz Varian/Oxford As-500 spectrometer, or an 850 MHz Bruker Avance III HD spectrometer. VT and low-temperature 2D-COSY ¹H NMR spectra were recorded on a 500 MHz Bruker Avance III spectrometer. Chemical shifts were referenced to the internal standard of tetramethylsilane (as $\delta$ = 0.00 ppm) or referenced to the residual solvent peaks. High-resolution electrospray ionization (ESI) mass spectra were obtained on an ESI-Q-TOF mass spectrometer (Compact, Bruker Daltonics Inc.). FT-IR spectra were recorded on a PerkinElmer Spectrum Two Fourier transform infrared (FT-IR) spectrometer. Elemental analysis was performed by a Perkin Elmer 2400 Series II CHNS/O Analyzer. MALDI-TOF spectra were recorded in positive-ion reflectron mode using an Applied Biosystems Voyager DE-STR.

**Computational studies.** All DFT[71] calculations were carried out using Gaussian 09 suite program[72]. Geometry optimizations and vibrational frequency calculations were conducted with B3LYP-D3 and 6–31 G(d,p) basis set[73,74]. For [**C-P2**]₂ and [**C-NI**]₂, crystallographically determined atomic coordinates were used as an input without a geometry optimization process. The calculated structures were varied by frequency calculations: no imaginary frequencies for the minima. Molecular electrostatic potential (MEP) maps were generated by using GaussView 5 and Gaussian 09 software.

## Data availability

The X-ray crystallographic coordinates for structures reported in this study have been deposited at the Cambridge Crystallographic Data Centre (CCDC), under deposition numbers 2015414 (**C-P2⊃PHD**), 2015415 (**C-NI**), 2015416 (**C-P1**), 2015417 (**C-NIMe**), 2015418 (**C-P2**), and 2015419 (**C-P4**). These data can be obtained free of charge from The Cambridge Crystallographic Data Centre via www.ccdc.cam.ac.uk/data_request/cif. A summary of X-ray crystallographic data is available in Supplementary Table 1.

Experimental procedures and supplementary figures are available in the Supplementary Information.

NMR spectra and DFT Cartesian coordinates are available in the Supplementary Data.

A schematic animation of self-assembly and fluxional motion is provided in the Supplementary Movie.

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

## Acknowledgements

This work was supported by the Samsung Science and Technology Foundation (SSTF-BA1701-10680), and the National Research Foundation of Korea (2020R1A2C2006381 and 2021R1A5A1030054).

## Author contributions

H.L., under the supervision of D.L., synthesized the molecules, designed the experiments, and performed spectroscopic and X-ray crystallographic measurements. Both authors contributed to the analysis of the results and the writing of the paper.

## Competing interests

The authors declare no competing interests.
