## [Peer Review File · Communications Chemistry]

REVIEWERS' COMMENTS:

Reviewer #1 (Remarks to the Author):

The paper by Lee et. al. describes the synthesis of molecular clips with strong hydrogen-bond interaction. The designed molecules contain pyridine as a hydrogen-bonding acceptor and an indole group as H-bonding donor. The platform can be ready to install with various pi-conjugated arms and tune the intermolecular pi-pi interaction. All synthesized complexes were well characterized with single crystal-XRD, MALDI-TOF-MS and NMR methods. The crystal structures provide solid support as the molecular clips. The dimerization process was fully evaluated with NMR techniques. Overall, this is nice work to create molecular clips and can be accepted under the current form.

Reviewer #2 (Remarks to the Author):

This paper submitted by Lee et al. describes the preparation of a four shape-persistent fully organic molecular clips and the detailed study of their self-assembly properties, affording discrete 'homo' dimers presenting quadruple pi-stacks or selected inclusion of pi-conjugated guest affording 'hetero' triple pi -stacks.

This studies first explores the syntheses, solid-state molecular structures and then investigates the solution self-assembly behaviors of these derivatives, varying the temperature, solvents and concentration together with submitting the newly prepared molecular clips to different planar pi-conjugated guest to eventually afford intercalated compounds. Titration experiments are also performed. Conclusive to these experiments, association constants are determined, compared and discussed. DFT calculations have been also carried out.

All in all, the synthesis, characterizations and experiments performed on this new derivative are well conducted. This is an interesting paper.

regarding the results reported, this well-written and clear article should, in my opinion, be published once the authors have considered the corrections suggested hereafter:

1) In the article's introduction, the synthesized molecular clips are defined as C-shaped ../.. to support two parallel-oriented pi-surfaces (lines 42-43). In my point of view, this choice is not suitable as a 'C'-shaped molecular clip more likely represents a pincer clip. The letter 'U'-shape should be rather preferred to design the targeted compounds geometry

2) Considering the introduction part, and the previous comment, the authors should mention the works that have been done using the coordination-driven supramolecular approach using pre-assembled molecular clips having a 'U'-shape (or 'C'-shape depending on the letter finally chosen) and for which similar cavities accepting pi- stacks have been obtained in the solid-state, as another source of inspiration. See for example:

- J. Am. Chem. Soc. 2003, 125, 46, 13950-13951
- Inorg. Chem. 2005, 44, 22, 7886-789
- J. Am. Chem. Soc. 2001, 123, 17, 3940-3952

Regarding the use of molecular clips having a 'U-shape' and leading to very similar assemblies and discrete or infinite stacks:

- Inorg. Chem. 2011, 50, 3183-3185.
- Chem. Eur. J., 2014, 20, 14853-14867
- Eur. J. Inorg. Chem. 2015, 2934-2938
- Acc.Chem. Res. 2017, 50 (4), 885-894

In addition, coordination-driven supramolecular chemistry also gave some examples of multiples stacks:

- J. Am. Chem. Soc. 2020, 142, 44, 18946–18954
- J. Am. Chem. Soc. 2004, 126, 48, 15852–15869

3) Figure 2 is not very easy to understand for the reaction affording the compounds 6x from 2x'. I would suggest the authors supply a scheme mostly organized in a horizontal way rather than vertically, using for example a 'big bracket' embracing both compounds 2x' and 4 from which an arrow with compound 5 will represent the reaction affording compounds 6x both from precursors 2x' and 4.

4) Line 118, the way interplanar distances have been measured should be mentioned (minimal distances between the mean plane of a fragment and atoms of the other fragment, mean distance between mean planes, ... ?)

5) Line 266, the solvent used for the DOSY experiment should be given in the main text since it is not clear considering several different solvents were used in the experiments described previously

6) Line 276: 'among various small molecule screened'. These molecules (mentioned in Figures S35 and S36 ?) should be named or given in a scheme.

7) How the curves given in figure 10 have been established? It is not clear considering the explanations given from lines 339 to 348.

8) The way the references are given is not homogeneous and should be corrected (Either the first author et al. or the full list of authors is given)

Reviewer #3 (Remarks to the Author):

H. Lee and D. Lee report on the synthesis and characterization of four different rigid molecular clips incorporating polyaromatic recognition units. The key feature of this family of molecules resides in the presence of a central N-H motif in the spacer enabling additional H-bond interactions to the expected π - π stacking. These novel systems form relatively stable dimers ($10^5 < K_a < 8 \times 10^5 \text{ mol L}^{-1}$) according to a quadruple π -stack arrangement. Finally, a competitive study was undertaken using 1,10-phenanthroline-5,6-dione. Data are fully consistent and the supporting information was performed with care. These systems were studied using an impressive number of techniques. In addition, the text is well-written and easy to read and figures are clear.

Minor comments:

1/ The field of the NMR spectrometer should be written in the caption of the related figures of the main text.

2/ Mass spectrometry is not a proof of dimers formation. Nonspecific aggregates can form in the gas phase. This confusion should be avoided in the main text as stated in page 5, last sentence.

These suggestions do not alter the quality of the manuscript and deserves to be published in Communications Chemistry.

Reviewer #1. The reviewer provided an excellent summary of our key discoveries, and concluded supportively that "Overall, this is nice work to create molecular clips and can be accepted under the current form."

Reviewer #2. The reviewer provided a detailed evaluation of our work, and commented that "This is an interesting paper. regarding the results reported, this well-written and clear article should, in my opinion, be published once the authors have considered the corrections suggested hereafter". Listed below are our point-by-point responses to her/his comments and helpful suggestions.

- 1) In the article's introduction, the synthesized molecular clips are defined as C-shaped ../. to support two parallel-oriented pi-surfaces (lines 42-43). In my point of view, this choice is not suitable as a 'C'-shaped molecular clip more likely represents a pincer clip. The letter 'U'-shape should be rather preferred to design the targeted compounds geometry.

▶ Analogies are always tricky. If taken too literally, they fail to achieve the intended goal of providing clarity or identifying similarities. Analogies of molecular shapes to everyday objects are no exception. Our molecular clips are designed with structural variations made in the "upper canopy" part attached to the common "L-shaped lower skeleton" part. With these two figurative descriptions for each structural subcomponent, the shape of the completed clip-like object is best described as the alphabet letter C, rather than U. In other words, the letter U in the upright position cannot be adequately deconstructed, at least conceptually, into the "upper canopy" and "L-shaped lower skeleton" parts, whereas the letter C can. This is the reason that we decided to use the letter C, instead of U.

▶ At the same time, we agree with the reviewer on that the letter U better describes the parallel-oriented nature of the two π -faces. A compromise is thus in need. We, therefore, rephrased the term "C-shaped" as "C-shaped (or horizontal lying letter U-shaped)" where it first appeared in the main text (Page 2, Paragraph 1, first sentence). We believe that this is an agreeable and reasonable solution.

- 2) Considering the introduction part, and the previous comment, the authors should mention the works that have been done using the coordination-driven supramolecular approach using pre-assembled molecular clips having a 'U'- shape (or 'C'-shape depending on the letter finally chosen) and for which similar cavities accepting pi- stacks have been obtained in the solid-state, as 1 another source of inspiration. See for example: - J. Am. Chem. Soc. 2003, 125, 46, 13950–13951 - Inorg. Chem. 2005, 44, 22, 7886–789 - J. Am. Chem. Soc. 2001, 123, 17, 3940–3952 Regarding the use of molecular clips having a 'U-shape' and leading to very similar assemblies and discrete or infinite stacks: - Inorg. Chem. 2011, 50, 3183-3185. - Chem. Eur. J., 2014, 20, 14853-14867 - Eur. J. Inorg. Chem. 2015, 2934–2938 - Acc.Chem. Res. 2017, 50 (4), 885–894 In addition, coordination-driven supramolecular chemistry also gave some examples of multiples stacks: - J. Am. Chem. Soc. 2020, 142, 44, 18946–18954 - J. Am. Chem. Soc. 2004, 126, 48, 15852–15869

► We thank the reviewer for bringing these works to our attention, from which the most relevant four articles are now cited in the revised manuscript (see below), and included as references #34, #42, #43, and #44:

"Prominent examples include aromatic guest-encapsulated molecular cages^{14,34–38}, through-space π -conjugated foldamers^{39–41}, metal coordination-induced π stacking^{42–44}, and self-assembled molecular tweezers^{45–49}." (Page 2, Paragraph 1)

34. Lin, R. et al. Self-assembly and molecular recognition of a luminescent gold rectangle. *J. Am. Chem. Soc.* 126, 15852–15869 (2004).

42. El Sayed Moussa, M. et al. Dissymmetrical U-Shaped π -stacked supramolecular assemblies by using a dinuclear CuI clip with organophosphorus ligands and monotopic fully π -conjugated ligands. *Chem. Eur. J.* 20, 14853–14867 (2014).

43. Attenberger, B. et al. Discrete polymetallic arrangements of AgI and CuI ions based on multiple bridging phosphane ligands and π - π interactions. *Eur. J. Inorg. Chem.* 2934–2938 (2015).

44. Lescop, C. Coordination-driven syntheses of compact supramolecular metallacycles toward extended metallo-organic stacked supramolecular assemblies. *Acc. Chem. Res.* 50, 885–894 (2017).

- 3) Figure 2 is not very easy to understand for the reaction affording the compounds 6x from 2x'. I would suggest the authors supply a scheme mostly organized in a horizontal way rather than vertically, using for example a 'big bracket' embracing both compounds 2x' and 4 from which an arrow with 2 compound 5 will represent the reaction affording compounds 6x both from precursors 2x' and 4.

► According to the reviewer's kind suggestion, revisions were made by expanding Fig. 2 into a two-column format, and reorganizing synthetic routes leading to the 6X-series molecules independently from the 2X-series (top) and 4 (bottom) precursors. The revised synthetic scheme now appears more intuitive to understand.

- 4) Line 118, the way interplanar distances have been measured should be mentioned (minimal distances between the mean plane of a fragment and atoms of the other fragment, mean distance between mean planes, . . . ?)

► The naphthaleneimide · · isobenzimidazole interplanar distance was determined by measuring the distance between the mean plane centroid of naphthaleneimide and the mean plane of pyridine–isobenzimidazole–pyridine unit. The isobenzimidazole · · isobenzimidazole

interplanar distance was determined by measuring the distance between the mean planes of the two stacked pyridine–isobenzimidazole–pyridine units.

► To make this point clear, the following sentences were added in the revised manuscript:

"These values were determined by measuring the distance between the mean plane centroid of naphthaleneimide and the mean plane centroid of pyridine–isobenzimidazole–pyridine unit (for a · · · b), and the mean centroids of the two stacked (and thus facing) pyridine–isobenzimidazole–pyridine triads (for b · · · b)." (Page 3, end of Paragraph 3)

- 5) Line 266, the solvent used for the DOSY experiment should be given in the main text since it is not clear considering several different solvents were used in the experiments described previously.

► According to the reviewer's suggestion, the solvent used for the ¹H DOSY experiments is reported in the revised manuscript, which reads " ¹H DOSY measurements on C-NI and C-Pn (n = 1, 2, and 4) in CDCl₃ . . ." (Page 6, toward the end of Paragraph 2) 3

- 6) Line 276: 'among various small molecule screened". These molecules (mentioned in Figures S35 and S36 ?) should be named or given in a scheme.

► To make this point clear, the text has been revised to read "Among various small molecules (see Supplementary Figs. 35 and 36 for a complete list) screened . . ." (Page 6, middle of Paragraph 3)

- 7) How the curves given in figure 10 have been established? It is not clear considering the explanations given from lines 339 to 348.

► The speciation plots in Fig. 10 were numerically simulated by using the experimentally determined dimerization constant ($K_{dim} = 1.3 \times 10^5 \text{ M}^{-1}$), host–guest complexation constant ($K_{HG} = 8.3 \times 10^3 \text{ M}^{-1}$), and a fixed total concentration of $[C-NI]_0 = 5.0 \text{ mM}$. The y-axis reports changes in the mole fractions of the free (= unbound) host C-NI, dimerized host $[C-NI]_2$, and host-guest complex $C-NI \supset PHD$ as a function of the total guest concentration $[PHD]_0$ corresponding to the x-axis.

► From eqs (2), (3), (4), and (7) in Supplementary Note 2, the following relationship is established: $[H] + 2K_{dim}[H]^2 + K_{HG}[H][G]_0 = [H]_0$ Using the K_{dim} , K_{HG} , and $[H]_0 (= [C-NI]_0)$ values as constants (see above), the concentration of the unbound host H (= [C-NI]) is calculated by varying the $[G]_0 (= [PHD]_0)$ value. The speciation plot (blue line in Fig. 10) is drawn for the change in the mole fraction $[C-NI]/[C-NI]_0 (= y\text{-axis})$ as a function of $[PHD]_0 (= x\text{-axis})$; as equiv amounts to $[C-NI]_0$.

► From eqs (2), (3), (4), and (7) in Supplementary Note 2, the following relationship is established: $s [H_2] K_{dim} + 2[H_2] + [G]_0 - \frac{[G]_0 K_{HG}}{r [H_2] K_{dim} + 1} = [H]_0$ Using the K_{dim} , K_{HG} , and $[H]_0 (= [C-NI]_0)$ values as constants (see above), the concentration of the dimeric $H_2 (= [C-NI]_2)$ is calculated by varying the $[G]_0 (= [PHD]_0)$ value. The speciation plot (red line in Fig. 10) is drawn for the change in the mole fraction $[C-NI]_2/[C-NI]_0 (= y\text{-axis})$ as a function of $[PHD]_0 (= x\text{-axis; as equiv amounts to } [C-NI]_0)$.

► From eqs (3), (5), and (7) in Supplementary Note 2, the following relationship is established: $K_{HG} p - 1 - 8K_{dim}([HG] - [H]_0) - 1 \cdot ([G]_0 - [HG]) = 4K_{dim}[HG]$ Using the K_{dim} , K_{HG} , and $[H]_0 (= [C-NI]_0)$ values as constants (see above), the concentration of the host-guest complex $HG (= [C-NI \supset PHD])$ is calculated by varying the $[G]_0 (= [PHD]_0)$ value. The speciation plot (green line in Fig. 10) is drawn for the change in the mole fraction $[C-NI \supset PHD]/[C-NI]_0 (= y\text{-axis})$ as a function of $[PHD]_0 (= x\text{-axis; as equiv amounts to } [C-NI]_0)$.

► The above information is now provided in the Supplementary Note 3 section of the revised Supplementary Information.

8) The way the references are given is not homogeneous and should be corrected (Either the first author et al. or the full list of authors is given).

► According to the journal (Nature) formatting guide, ". . . all authors should be included in reference lists unless there are more than five, in which case only the first author should be given, followed by 'et al.'. We strictly followed this rule.

Reviewer #3. The reviewer provided a concise summary of our key findings, and commented that "Data are fully consistent and the supporting information was performed with care. These systems were studied using an impressive number of techniques. In addition, the text is wellwritten and easy to read and figures are clear." Listed below are our point-by-point responses to her/his comments and helpful suggestions.

1/ The field of the NMR spectrometer should be written in the caption of the related figures of the main text.

► According to the reviewer's suggestion, figure captions of Fig. 3, Fig. 5, Fig. 6, and Fig. 9 have been revised to report the field of the NMR spectrometer used.

2/ Mass spectrometry is not a proof of dimers formation. Nonspecific aggregates can form in the gas phase. This confusion should be avoided in the main text as stated in page 5, last sentence.

► This is a legitimate concern which we fully agree with. The sentence that the reviewer specifically pointed out has been revised to read: "Unlike C-NI (Fig. S3a), the MALDI-TOF mass spectrum of C-NIME (Fig. S3e) lacks the molecular ion peak corresponding to the dimeric $[2M + H]^+$ species." (Page 3, end of

Paragraph 4) By doing so, we simply report experimental observations, not making any speculative claims.